

# Review of models for estimating 3D human pose using deep learning

Sani Salisu[1], Kamaluddeen Usman Danyaro[2], Maged Nasser[2], Israa M. Hayder[3] and Hussain A. Younis[4]

[1] Department of Information Technology, Federal University Dutse, Dutse, Jigawa, Nigeria
[2] Computer & Information Sciences Department, Universiti Teknologi PETRONAS, Seri Iskandar, Malaysia
[3] Department of Computer Systems Techniques, Qurna Technique Institute, Southern Technical University, Basrah, Iraq
[4] College of Education for Women, University of Basrah, Basrah, Iraq

## ABSTRACT

Human pose estimation (HPE) is designed to detect and localize various parts of the human body and represent them as a kinematic structure based on input data like images and videos. Three-dimensional (3D) HPE involves determining the positions of articulated joints in 3D space. Given its wide-ranging applications, HPE has become one of the fastest-growing areas in computer vision and artificial intelligence. This review highlights the latest advances in 3D deep-learning-based HPE models, addressing the major challenges such as accuracy, real-time performance, and data constraints. We assess the most widely used datasets and evaluation metrics, providing a comparison of leading algorithms in terms of precision and computational efficiency in tabular form. The review identifies key applications of HPE in industries like healthcare, security, and entertainment. Our findings suggest that while deep learning models have made significant strides, challenges in handling occlusion, real-time estimation, and generalization remain. This study also outlines future research directions, offering a roadmap for both new and experienced researchers to further develop 3D HPE models using deep learning.

## INTRODUCTION

Human pose estimation (HPE) means identifying the pose of human body segments and key points or joints in images, videos, and real-time environments. It involves tracking, detecting, and grouping the semantic key points of a given object for solving human problems such as clinical and rehabilitation solutions among others. HPE offers geometric and motion information about the human body which has been applied to a wide variety of applications such as human-computer interaction, motion analysis, augmented reality (AR), virtual reality (VR), healthcare, action recognition, animation, *etc*… and is used in several areas of human endeavours which include gaming industries, movies industries, entertainments, academic, professional research, *etc*. Different methods have been proposed for 3D human pose estimation, including silhouette contours (*Mondal, Ghosh & Ghosh, 2013*), edge-based histograms (*Mori & Malik, 2002*), pictorial structures (PS) (*Andriluka, Roth & Schiele, 2009*; *Dantone et al., 2013*) and deformable part models

Corresponding author
Sani Salisu, sani.salisu@fud.edu.ng

(DPMs) (*Felzenszwalb et al., 2010*), continued to build appearance models for each key points separately. Due to the features complexity of 3D human pose estimation such as viewpoint invariant 3D feature maps (*Haque et al., 2016*), histograms of 3D joint locations, multifractal spectrum, and volumetric attention models, traditional models (*Barajas, Dávalos-Viveros & Gordillo, 2013*; *Chan, Koh & Lee, 2013*; *Dinh et al., 2013*; *Handrich & Al-Hamadi, 2013*; *Xul et al., 2013*; *Belagiannis et al., 2014*; *Jung & Kim, 2014*; *Liang et al., 2014*; *Zhu et al., 2014*) are not able to perform accurately in HPE research.

Despite the achievement in solving issues related to pose estimation, HPE methods are facing challenges in detecting, capturing, and extracting the significant key points of the human body. Such challenges include occlusion (self-occlusion, inter-person occlusion, and out-of-frame occlusion), limited data (limited annotation, limited variation of pose, limited number of pose), bad input data (blurry, low resolution, low light, low contrast, small scale, noisy), domain gap, camera-centric, crowd scenes, speed, complex pose, *etc.* HPE has recently attracted increasing attention in the computer vision community in facing those challenges. A large number of deep learning-based models have been developed by enhancing the existing model to deal with those challenges facing both 2D (*Li et al., 2021*; *Liu et al., 2022*) and 3D (*Saini et al., 2022*; *Wu et al., 2022*) pose estimation. An article review is one of the most significant and effective approaches for guiding future researchers about the state-of-the-art of any scientific domain.

However, most of the existing surveys and literature reviews on human pose estimation focused on 2D HPE (*Jingtian et al., 2020*; *Munea et al., 2020*; *Ulku & Akagündüz, 2022*) while survey and literature reviews on 3D models HPE are limited. Out of the few surveys and comprehensive literature reviews on 3D HPE (*Ji et al., 2020*; *Wang et al., 2021*; *Toshpulatov et al., 2022*; *Tian et al., 2023*; *Azam & Desai, 2024*), none of these focus on the progress of the state-of-the-art deep learning-based 3D human pose estimation models. For example, *Azam & Desai (2024)* aim to estimate human body poses and develop body representations from a first-person camera perspective, focusing solely on egocentric human pose estimation and neglecting other 3D pose estimation issues. *Ji et al. (2020)* concentrates on monocular 3D images, disregarding those captured using binocular or stereo vision systems. *Zhang et al. (2021)* extensively reviews deep learning supervision models but only covers research articles from 2013 to 2021, missing recent advancements. A general survey of both 2D and 3D human pose estimation, including classical and deep learning approaches, is presented (*Dubey & Dixit, 2023*), making it challenging for researchers to find specific methodologies. *Wang et al. (2021)* published a review on 3D deep learning-based human estimation, but it heavily emphasizes 3D pose estimation datasets Furthermore, the literature reviews some existing 3D HPE researchers and presents a description of their methods and model architectures.

## RELATED WORK

*Wang et al. (2023)* reviews deep learning methods for 3D pose estimation, summarizes their pros and cons, and examines benchmark datasets for comparison and analysis, offering insights to guide future model and algorithm designs. A novel method to extract 3D information from 2D images without 3D pose supervision using 2D pose annotations

and perspective knowledge to generate relative depth of joints was proposed (*Qiu et al., 2023*). The authors introduced a 2D pose dataset (MCPC) and a weakly-supervised pre-training (WSP) strategy for depth prediction. WSP improves depth prediction and generalization for 3D human pose estimation, achieving state-of-the-art results on benchmark datasets. In *Pavlakos et al. (2019)* a 3D model of body pose, hand pose and facial expression from a single monocular image using Skinned Multi-Person Linear (SMPL)-X was computed. The authors improved upon the SMPLify approach by detecting 2D features for the face, hands, and feet, training a new neural network pose prior, defining a fast interpenetration penalty, and automatically detecting gender. Their newly implemented SMPLify-X significantly speeds up fitting SMPL-X to images. *Sun et al. (2020)* address the issue of monocular 3D human pose estimation with deep learning. To overcome occlusion problems inherent in single-view methods, they proposed an end-to-end network that generates multi-view 2D poses from single-view 2D poses, uses data augmentation for multi-view 2D pose annotations, and employs a graph convolutional network to infer 3D poses.

*Clemente et al. (2024)* explores the feasibility of a model for 3D HPE from monocular 2D videos (MediaPipe Pose) in a physiotherapy context, by comparing its performance to ground truth measurements. MediaPipe Pose was investigated in eight exercises typically performed in musculoskeletal physiotherapy sessions, where the range of motion (ROM) of the human joints was the evaluated parameter. This model showed the best performance for shoulder abduction, shoulder press, elbow flexion, and squat exercises. Results have shown that their model has achieved a higher performance. *He et al. (2024)* proposed a novel approach for telerehabilitation based on deep learning 3D human pose estimation. Their approach aims to evaluate the effectiveness and practicality of the telerehabilitation method over a 12-week experiment through a randomized controlled trial on older adults with sarcopenia, this study compared the training effects of an AI-based remote training group using deep learning-based 3D human pose estimation technology with those of a face-to-face traditional training group and a general remote training group.

## HUMAN BODY MODELLING

Because humans vary in their shapes and sizes, modelling the human body is a crucial aspect of HPE. To determine the posture of an individual, their body needs to meet the specific criteria necessary for a particular task to establish and describe the human body's pose. HPE frequently employs five distinct categories of human body models that include the kinematic-base model, planner model, volumetric model, SMPL-based model and Surface-based model (*Wang et al., 2021*; *Salisu et al., 2023*).

### Skeletal-based model

This model is commonly referred to as a kinematic-based model or a stick figure, characterized by its uncomplicated and adaptable representation of the human body's structure. It finds frequent application in both 2D (*Cao et al., 2017*) and 3D HPE (*Mehta et al., 2018*). This model primarily captures the positions of joints and limb orientations to depict the human body's skeletal structure, facilitating the detection of connections

between different body parts. This human skeleton model is conceptualized as a tree-like structure, encompassing numerous key points in the human body. It establishes connections between neighbouring joints through edges. The fusion of a convolutional neural network pose regressor and kinematic skeletal fitting allows for the real-time capture of a comprehensive 3D skeletal pose in a given environment using only a single RGB camera (*Terreran, Barcellona & Ghidoni, 2023*). While the kinematic model offers flexibility in graph representation, it has constraints in effectively representing texture and shape information.

## Contour-based model

The contour-model, also known as the planer model stands in contrast to the kinematic-based model. In the contour-based model, essential points are approximated with rectangular shapes or object boundaries. This model is primarily employed to represent the outline and structure of the human body. The planar model is a common choice in classical HPE methods (*Jiang, 2010*), such as those that utilize techniques like cardboard (*Freifeld et al., 2010*) mode and active shape modelling to capture the human body's structure and silhouette distortions through principal component analysis (PCA). In this model, researchers commonly depict body parts using rectangles that approximate the contours of the human body.

Many scholars employ this model to address issues related to the relationships between various human body parts. For instance, *Toshpulatov et al. (2022)* applied the planar model to capture the connections between different body parts, and their findings indicate that this model effectively represents the shape and appearance of the human body. *Ju, Black & Yacoob (1996)* proposed that an individual can be portrayed as a collection of interconnected planar patches. Their study demonstrated that limbs can be represented by these planar patches, offering a useful approach for tracking human legs across extended image sequences. Additionally, *Black & Yacoob (1995)* illustrated that a planar model can accurately approximate the motion of a human head, providing a succinct depiction of optical flow within a specific region.

## SMPL-based model

The Skinned Multi-Person Linear (SMPL) model, as introduced by *Loper et al. (2015)*, serves as a tool for predicting 3D human body joint locations, as described by *Bogo et al. (2016)*. This model represents the human skin as a mesh with 6,890 vertices, which can be adjusted through shape and pose parameters. Shape parameters are employed to capture aspects like body proportions, height, and weight, while pose parameters account for the specific deformations of the body. By learning and optimizing these shape and body parameters, one can estimate the 3D positions of the body's pose. Several researchers attempted to address the issue of SMPL-based by utilising single or combined models.

Other researchers shifted their attention to an enhanced version of the SMPL model, recognizing its limitations, such as computational complexity and the absence of facial and hand landmarks. Some of these researchers sought to overcome these constraints. For instance, *Xiu et al. (2022)* introduced an iterative refinement of SMPL parameters during

3D reconstruction. Similarly, *Lin, Wang & Liu (2021a*, *2021b)* put forward as methods for reconstructing 3D human pose and mesh from a single image, without depending on any parametric mesh model like SMPL.

### Surface-based model

A more recent human body model known as DensePose proposed by *Güler, Neverova & Kokkinos (2018)* has been introduced in response to the limitations of sparse image key points in comprehensively describing the human body's condition. To overcome this limitation, a fresh dataset called DensePose-COCO has been created. This dataset establishes detailed correspondences between image pixels and a surface-based representation of the human body, enhancing the ability to capture the human body's state.

### Volumetric model

Volume-based models are employed to depict the silhouette and pose of a three-dimensional object using a geometric mesh. Traditional geometric meshes used for modelling human body parts included shapes like cylinders and cones. In contrast, modern volume-based models are recognised by their mesh representations derived from 3D scans. These models represent the body as a 3D volume, often using voxels (3D pixels) or implicit functions to capture both the outer shape and potential internal structures. Volumetric models give a complete spatial representation of the body (*Salisu et al., 2023*). Among the most widely used volumetric models for 3D pose estimation are the Stitched Puppet Model (SPM) and the Unified Deformation Model (UDM) (*Trumble et al., 2017*), as well as models like Frankenstein & Adam and the Generic Human Model (GHUM) along with the Low-Resolution Generic Human Model (GHUML) (*Xu et al., 2020*).

## METHODOLOGY

On the topic of "3D human pose estimation using deep learning," an extensive study was conducted, investigating key research questions and systematically searching and organizing the relevant literature. Our research methodology adopted the Preferred Reporting Items for Systematic Reviews and Meta-Analyses (PRISMA) protocol (*Moher et al., 2009*) to outline the data sources, search strategies, and criteria for literature inclusion.

### Research questions

The survey articles aim to address the following questions

RQ1 How 3D human pose estimation evolved and developed.
RQ2 How are neural networks applied to different 3D pose estimation task.
RQ3 What areas of human endeavour where human pose estimation is applicable.
RQ4 What are the current challenges of 3D HPE.
RQ5 What are the future research directions for 3D human pose estimation.

## Research strategy and data sources

This review utilized automatic and manual search methods to ensure optimal outcomes. Four databases (Scopus, Web of Science, Google Scholar, and IEEE) were employed to locate pertinent articles or research within the 3D human pose estimation (3DHPE) domain and its applications. Specific search queries, comprising various keywords and their combinations were employed to identify relevant publications from 2014 to 2024. These queries included terms such as "3D human pose estimation," "3D human body model," "3D datasets and evaluation metrics," "3D HPE application," and "deep-learning based 3DHPE." To ensure impartial coverage, identical queries were executed across all four databases.

## Inclusion/exclusion criteria

Given the objective of conducting a comprehensive review tailored to the study's requirements, slight variations in search strategies were adopted, considering the unique search capabilities of each selected database. Initially, title and abstract searches were conducted in IEEE and Scopus, whereas full-text searches were performed in Web of Science and Google Scholar. Retrieved articles underwent scrutiny based on their abstracts and titles to determine inclusion or exclusion eligibility. Articles lacking sufficient or relevant information were excluded.

Subsequently, the full text of screened papers was meticulously examined to ascertain their relevance for inclusion or exclusion. Additionally, references cited within selected articles were identified and utilized to retrieve additional relevant papers for the study. To ensure the generation of clean and standardized documents, devoid of noise and duplicates, supplementary selection and rejection criteria were applied. These criteria stipulated that articles must be written in English, and published in English journals or conferences between the years 2014-2024. Furthermore, the articles were required to employ deep-learning techniques rather than classical methods.

## RESULTS

The literature search identified 601 articles on 3D human pose estimation. After removing 299 duplicates, 302 unique articles remained. Screening the titles and abstracts reduced this number to 97. Following a full-text review, 67 additional articles were excluded, leaving 30 relevant studies for inclusion in the review. Figure 1 illustrates the stages of the search from the input query, the inclusion and exclusion criteria to the final inclusion stage. A total of three studies ($n = 3$) were published in 2024, and eight studies ($n = 8$) were published in 2023. The remaining studies were distributed as follows: seven ($n = 7$) in 2022, six ($n = 6$) in 2021, three ($n = 3$) in 2020, two ($n = 2$) in 2019, and one ($n = 1$) in 2017. Figure 2 illustrates the growing research interest in the field of 3D human pose estimation (3DHPE). A critical evaluation was conducted on all 30 contributions that met the filtering criteria. In 3DHPE research, the choice of dataset plays a crucial role. Some researchers used benchmark datasets (*Yin, Lv & Shao, 2023*), while others employed self-prepared datasets (*Niu et al., 2024*; *Yan et al., 2024*) or a combination of both

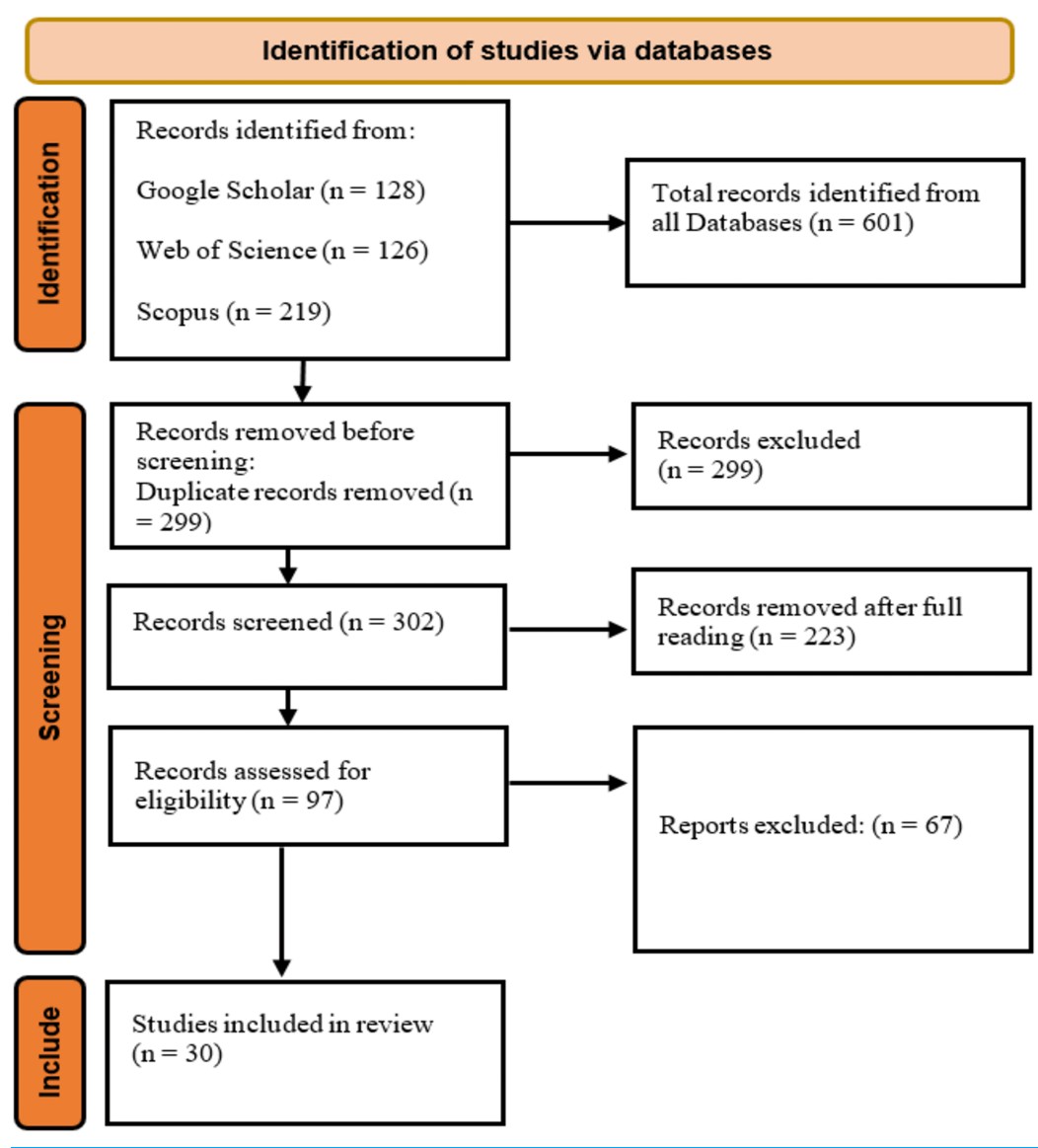

**Figure 1 Article selection process using PRISMA protocol.** Direct component sources. https://creativecommons.org/licenses/by/4.0/.               

(*Xi et al., 2024*). Human 3.6M emerged as the most frequently used dataset among the identified studies. A summary of the critical evaluation of all 30 contributions is presented in Table 1.

**RQ1: HOW 3D HUMAN POSE ESTIMATION EVOLVED AND DEVELOPED**

The fundamental process of HPE comprises three main phases. Firstly, it involves identifying key points/joints in the human body, such as the knee, ankle, shoulder, head, arms, and hands. This initial stage is crucial for pinpointing the specific locations of these key points. The choice of human pose dataset format plays a significant role in gathering and recognizing key points stored in selected 2D datasets. It is important to note that the

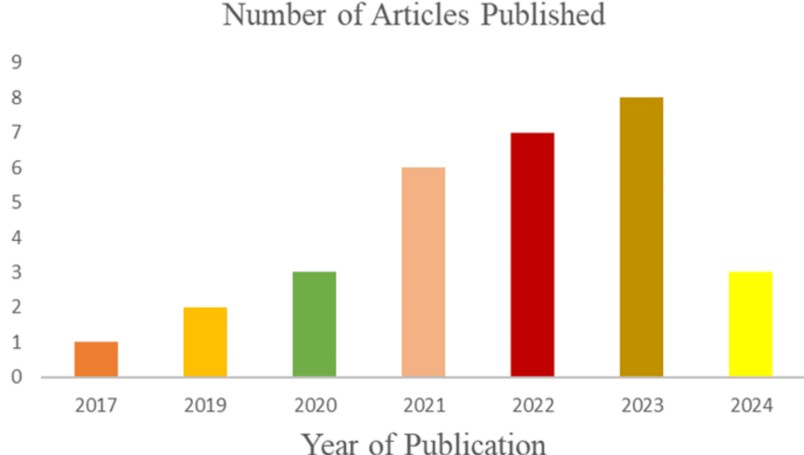

**Figure 2** **The number of published articles from 2017 to 2024.** The growing research interest in the field of 3D human pose estimation (3DHPE).

**Table 1** **The summary of the current deep learning-based article from Scopus.** Summarizes the literature of 30 of the most recently published articles in Scopus, illustrating the study, the model utilized, the type of dataset and a summary of the aim, achievement made and the limitation of each study.

| Study | Model | Dataset | Comments |
|---|---|---|---|
| Xi et al. (2024) | TCN, T-MHSA | Human 3.6M and MPI-INF-3DHP | **Aim:** to enhance the precision of pose estimation. |
| | | | **Achievement:** effectiveness in handling real-time and complex sports scenarios. |
| | | | **Limitation:** scalability, the robustness of the approach across various sports. |
| Zhang et al. (2023c) | P2P-MeshNet | Self-prepared (FreeMocap) | **Aim:** to estimate the body joint rotations directly |
| | | | **Achievement:** potential application prospects of the method |
| | | | **Limitation:** Could be further evaluated on different datasets for generalisation. |
| Yan et al. (2024) | IP +DL + EA | Self-prepared | **Aim:** Provide a reliable and efficient solution for 3D human pose estimation and awkward posture detection in a construction environment. |
| | | | **Achievement:** Contributes to the proactive management of worker health and safety. |
| | | | **Limitation:** Not applicable to health care monitoring data |
| Martini et al. (2022) | MAEVE platform | MSCOCO | **Aim:** Implement a low-cost, real-time and usable platform for 3DPE that guarantees accuracy. |
| | | | **Achievement:** Real-time performance and higher accuracy. |
| | | | **Limitation:** Future improvements could focus on enhancing adaptability across various HPE scenarios and patient populations to strengthen its clinical utility |
| Ran et al. (2023) | De-occlusion multi-task learning network | 3DPW, 3DPW-OCC, 3DOH, and Human 3.6 M | **Aim:** To de-noise the feature for mesh parameter regression |
| | | | **Achievement:** Competitive performance on a non-occlusion dataset |
| | | | **Limitation:** Real-world applicability and processing speed remain potential areas for future work |

| Study | Model | Dataset | Comments |
|---|---|---|---|
| Vukicevic et al. (2021) | VIBE | Self-prepared | **Aim:** To utilize IoT force sensors and IP cameras to detect unsafe P&P acts timely and objectively. |
| | | | **Achievement:** Excellent covenant with motion sensors and a high potential for checking and improving the safety of the P&P workplace. |
| | | | **Limitation:** Not applicable to P&P datasets |
| Bigalke et al. (2023) | Domain adaptation | Self-prepared and SLP dataset | **Aim:** Implement a model from a labeled source to a shifted unlabeled target domain |
| | | | **Achievement:** Outperformed the SOTA method (baseline and gap) |
| | | | **Limitation:** Tested on SLP and MVIBP datasets only |
| Yin, Lv & Shao (2023) | COG | Human 3.6M | **Aim:** To design a multi-branch network based on the human center of gravity |
| | | | **Achievement:** higher efficiency and validity |
| | | | **Limitation:** Need further exploration into integrating additional contextual features |
| Xu et al. (2022) | RSC-Net | Human 3.6M and MPI-INF-3DHP | **Aim:** To implement a method that can deal with the issues of low-resolution input images or video. |
| | | | **Achievement:** Accurate learning of 3D body pose and shape across different resolutions with one single model. |
| | | | **Limitation:** Future work might focus on further refining the robustness of cross-view matching in complex real-world settings. |
| Mehta et al. (2017) | VNect | MPI-INF-3DHP and Human 3.6M | **Aim:** To develop a real-time model that captures the full body 3D skeletal human pose in a steady, temporally reliable manner using a single RGB camera. |
| | | | **Achievement:** Better applicability than RGB-D solutions. |
| | | | **Limitation:** Not capable of estimating 3D poses from different camera views |
| Li et al. (2020a) | Dual–Stage pipeline | 3D human pose datasets, Human 3.6M and MPI INF-3DHP. | **Aim:** To develop a self-supervised model to avoid manual annotations of 3D poses. |
| | | | **Achievement:** more effectiveness compared to the current weekly-supervised model. |
| | | | **Limitation:** Relying on the completeness of the shape prior provided by RGBD-PIFu. |
| Saini et al. (2022) | UAVs + AirPose | Self-prepared | **Aim:** To develop a new model that estimates human pose and shape using images captured by multiple irrelevantly uncelebrated flying cameras |
| | | | **Achievement:** 3D HPE system for unstructured, uncontrolled and outdoor environments |
| | | | **Limitation:** Not robust to various lighting conditions and diversifying training data. |
| Yang et al. (2022) | PoseMoNet | Human 3.6M and HumanEva-I | **Aim:** To develop an elf-projection mechanism that cogently conserves human motion kinematics. |
| | | | **Achievement:** A competitive advantage compared to SOTA |
| | | | **Limitation:** Limited mechanisms to handle more varied and complex motions. |
| Gao, Yang & Li (2022) | GroupPoseNet | InterHands2.6M | **Aim:** To develop a 3D PE model that can differentiate between two hand shapes and poses from a single RGB image. |
| | | | **Achievement:** Higher efficiency in the overall result. |
| | | | **Limitation:** Lack of generalization capabilities for the real-world environment. |

(Continued)

| Study | Model | Dataset | Comments |
|---|---|---|---|
| *Kourbane & Genc (2022)* | Two-stage GCN-based | STB and RHD | **Aim:** To develop a model that learns per-pose relationship constraints in estimating 3D hand pose.<br>**Achievement:** Outperforms SOTA in terms of accurate 3D hand pose estimation.<br>**Limitation:** Future research could explore optimizing computational demands and validating model generalizability across more diverse, real-world scenarios. |
| *Pavlakos et al. (2019)* | SMPLify-X | EHF and SMPL+H | **Aim:** Simplify the analysis of human actions and emotions estimation.<br>**Achievement:** Higher speed than the SOTA methods.<br>**Limitation:** lack of a dataset of in-the-wild SMPL-X fits, which restricts the current ability to learn a regressor that can directly estimate SMPL-X parameters from RGB images. |
| *Mehrizi et al. (2019)* | DNN + Hourglass network | Self-prepared | **Aim:** This study aims to develop and validate a DNN-based model for 3D pose estimation during lifting<br>**Achievement:** Higher accuracy even with the shortcomings of marker-based motion systems.<br>**Limitation:** The number and position of cameras were not explored, the study focused on lifting tasks only and markers on the body may alter the natural appearance of the body. |
| *Zhu et al. (2023)* | MHPT | MoVi | **Aim:** To develop a deep-learning human pose technique for clinical gait analysis.<br>**Achievement:** pose estimations have been improved significantly.<br>**Limitation:** The system needs further validation across diverse patient groups and conditions to assess its reliability fully in clinical practice. |
| *Kong & Kang (2021)* | 3DMPPE | Human 3.6M | **Aim:** To develop a model to reduce computation cost and processing time.<br>**Achievement:** Reducing the processing time and the performance of the model.<br>**Limitation:** Lack of optimisation and adaptability to diverse scenarios. |
| *Zou et al. (2021)* | EventHPE | Self-prepared and DHP19 | **Aim:** To develop a stage deep learning model for accurate pose estimation.<br>**Achievement:** Effectiveness of the new model.<br>**Limitation:** Could enhance its real-time performance and adaptability to diverse environments. |
| *Šajina & Ivašić-Kos (2022)* | Tracking | Self-prepared | **Aim:** To develop a new model, that will enhance the 3D sequences of poses generated from the designed testing model.<br>**Achievement:** Revealing the drawbacks of other methods in the field of HPE.<br>**Limitation:** Lacks of advanced occlusion handling techniques to improve reliability across various sports contexts. |
| *Zhang et al. (2023a)* | PoseAug | Human 3.6M | **Aim:** To develop a model that will cater for the variation of 2D and 3D pose pairs.<br>**Achievement:** higher improvement on both frame and video-based 3D HPE.<br>**Limitation:** The model needs Further testing to confirm its generalizability. |
| *Zhou, Dong & EI Saddik (2020)* | DGCNN | ITOP and EVAL | **Aim:** To develop a model that will solve the issues of 3DHPE using depth images.<br>**Achievement:** Higher accuracy than SOTA models<br>**Limitation:** Not applicable to a fast and occluded movement. |

| Study | Model | Dataset | Comments |
|---|---|---|---|
| *Retsinas, Efthymiou & Maragos (2023)* | Template mushroom model | Self-prepared | **Aim:** To develop a deep learning model that solves the annotation problem and estimates their pose on 3D data. |
| | | | **Achievement:** more effectiveness compared to SOTA models. |
| | | | **Limitation:** Further research could enhance the approach's robustness through domain adaptation and extensive testing on real-world data. |
| *Ding et al. (2021)* | Kinematic Constrained Learning | Self-prepared | **Aim:** To develop a learning model for predicting skeletal key points from observed radar data. |
| | | | **Achievement:** The fusion of kinematic constraints with the learning of 3D skeletal reconstruction. |
| | | | **Limitation:** The model's performance should be validated in varied environments |
| *Ying & Zhao (2021)* | 3D learning module | MHAD and SURREAL | **Aim:** To develop a model that can estimate 3D human pose from RGB-D images. |
| | | | **Achievement:** SOTA performance on the stated datasets. |
| | | | **Limitation:** further validation in diverse settings and improvements in computational efficiency would be necessary for broader real-world applicability. |
| *Niu et al. (2024)* | SPCT+TRP | Self-prepared | **Aim:** To attack the intrinsic difficulties of the classical model |
| | | | **Achievement:** Improves accuracy and robustness in challenging conditions. |
| | | | **Limitation:** Lack of broader testing and validation across different settings |
| *Rapczyński et al. (2021)* | Scale normalization+ OpenPose | HumanEva-I, Human 3.6M, and Panoptic Studio, | **Aim:** To develop a model that will tackle the issue of both cross and dataset generalisation. |
| | | | **Achievement:** Improvements in cross-dataset and in-dataset generalisation. |
| | | | **Limitation:** Manual parameterisation for each new dataset. |
| *Manesco, Berretti & Marana (2023)* | Domain Unified Approach | SURREAL and Human 3.6M | **Aim:** solving pose misalignment problems on a cross-dataset scenario. |
| | | | **Achievement:** showing significant improvements in cross-domain accuracy by leveraging the domain adaptation technique |
| | | | **Limitation:** Future directions might involve testing across more diverse datasets and addressing computational efficiency for real-time use |
| *Sun et al. (2020)* | End-to-end 3DPEN | Human 3.6M | **Aim:** to address the issue of not using single-view images directly in multi-view methods. |
| | | | **Achievement:** higher effectiveness and performance improvement |
| | | | **Limitation:** Not validated on complex poses and diverse datasets. |

output of body key points from the same image may differ based on the dataset format and platform employed.

Moving on to the second stage, pose estimation entails grouping the localized key points to form valid human pose configurations, thereby determining pairs of organs in the human body. Various researchers have experimented with different techniques for connecting key point candidates in this stage.

The third and final stage involves estimating a 3D pose based on the previously determined 2D key points. This is achieved by combining sample 2D frames captured at

different times through a temporal procedure known as a temporal convolution neural network.

## 3D human pose estimation

In recent years, machine learning approaches have significantly transformed pose estimation, with deep learning (DL) methods making notable strides in enhancing its performance. While substantial progress has been made in 2D HPE, the task of 3D HPE remains challenging. Many existing studies address 3D HPE using monocular images or videos, presenting an ill-posed and inverse problem due to the loss of one dimension in the projection from 2D to 3D. However, when multiple views or additional sensors like Inertial Measuring Units (IMU) are utilized, 3D HPE becomes a well-posed problem that can benefit from information fusion techniques. Below is the description of 3D HPE using various sensors, including camera sensors, IMU sensors, point cloud and depth sensors, and radiofrequency device sensors.

### 3D HPE from digital RGB images and videos

3D HPE using deep learning from digital RGB images and video can be classified into three categories: single-view single-person 3D HPE, single-view multi-person 3D HPE, and multi-view 3D HPE.

A single-view single-person 3D HPE: Methods for single-person 3D HPE can be categorized into two groups, namely, skeleton-only and human mesh recovery (HMR). This classification is based on whether the goal is to reconstruct a 3D human skeleton or to recover a 3D human mesh using a human body model.

Skeleton-only: Methods falling into the skeleton-only category produce 3D human joint estimates as their ultimate output. These approaches do not utilize human body models for reconstructing a 3D human mesh representation. Within this category, these methods can be further subcategorized into direct estimation approaches and 2D to 3D lifting approaches.

#### Regression-based estimation

Regression-based estimation methods deduce the 3D human pose directly from 2D images, without the need for an intermediate step of estimating a 2D pose representation. This study (*Liang, Sun & Wei, 2018*) introduced a regression approach that considers the data's structure. Instead of relying on a joint-based representation, they opted for a more stable bone-based description. They defined a compositional loss by leveraging the 3D bone structure, using the bone-based description to encode long-range relations between the bones. A volumetric approach to transform the challenging non-linear 3D coordinate regression task into a handier form within a discretized space was presented (*Pavlakos et al., 2017*; *Pavlakos, Zhou & Daniilidis, 2018*). A convolutional network was employed to predict voxel likelihoods for each joint in the volume. They utilized ordinal depth relations of human joints to mitigate the requirement for precise 3D ground truth pose information.

*2D to 3Dlifting*

Inspired by the recent achievements in 2D human pose estimation (HPE), the popularity of 2D to 3D lifting approaches has risen. These methods involve inferring the 3D human pose from an interim estimation of the 2D human pose. During the initial phase, preexisting 2D HPE models are utilized to predict the 2D pose. Subsequently, in the second stage, the process of 2D to 3D lifting is applied to derive a 3D pose. A fully connected residual network for the regression of 3D joint locations, relying on the provided 2D joint locations was introduced by *Tralic et al. (2013)*. While this method achieved state-of-the-art results during its time, its susceptibility to failure stemmed from the reconstruction ambiguity resulting from excessive dependence on the 2D pose detector. The most optimal 3D pose was identified in *Jahangiri & Yuille (2017)*, *Li & Lee (2019)*, *Sharma et al. (2019)* by initially producing various 3D pose possibilities and subsequently utilising ranking networks to optimise the output.

## Human body recovery

Human body recovery (HBR) methods also known as human mesh recovery integrate parametric body models, as outlined in Section 2 (human body modelling), to reconstruct the human mesh. The 3D pose is then acquired by utilizing the joint regression matrix defined by the model. Common examples of HBR are SMPL-based volumetric-based and surface-based models.

### Single-view multi-person 3D HPE

Multi-person 3D HPE from monocular RGB images or videos has a more compound and demanding challenge, involving the identification of the number of individuals, their positions, and poses, followed by the grouping of their localized body key points and finally estimating the 3D pose. To address these challenges, multi-person pose estimation can be categorized into two techniques: Top-down and Bottom-up.

*Top-down techniques*

Top-down approaches in 3D multi-person HPE initially engage in human detection to identify each person. Subsequently, for every detected individual, the absolute root coordinate and 3D root-relative pose are estimated through 3D pose networks. This process involves inputting the image into the human detection network, cropping the detected humans, aligning them to the world coordinate, and finally estimating the 3D root-relative pose.

Some researchers like *Zou et al. (2023)* introduce Snipper, a cohesive framework designed to execute multi-person 3D pose estimation, tracking, and motion forecasting concomitantly within a single stage. Their approach incorporates a proficient yet robust deformable attention mechanism, enabling the aggregation of spatiotemporal information from the video snippet. Leveraging this deformable attention mechanism, they train a video transformer to capture spatiotemporal features from the multi-frame snippet and generate informative pose features for multi-person pose queries. Ultimately, these pose queries are processed to predict both multi-person pose trajectories and future motions in a single shot.

*Bottom-up techniques*

Differing from top-down strategies, bottom-up approaches initially generate the locations of all body joints and depth maps. Subsequently, these methods associate body parts with each individual based on the root depth and relative depth of each part. Grouping the human body joints belonging to each person is one of the major challenges of the bottom-up technique. For example, *Xiao et al. (2023)* presented a refined method for body representation and a streamlined single-stage multi-person pose regression network named AdaptivePose++. The innovative body representation can adequately capture diverse pose information and efficiently model the connection between a human instance and its associated key points within a single forward pass.

### Multi-view multi-person 3D HPE

Multi-view 3D HPE encounters difficulties in handling partial occlusion when operating in a single-view setting. A viable approach to address this challenge involves estimating the 3D human pose from various viewpoints. This is because the obscured portions in one view might be visible in other views. To achieve 3D pose reconstruction from multiple perspectives, it is crucial to resolve the association of corresponding locations across different cameras. Many researchers attempted to overcome the issue of partial occlusion associated with single-view 3D HPE (*Wang et al., 2021b*; *Liu, Wu & He, 2022*; *Xu & Kitani, 2022*; *Gerats, Wolterink & Broeders, 2023*; *Silva et al., 2023*). Generally, multi-view settings are primarily employed for multi-person pose estimation.

## 3D HPE from digital image with imu sensors

Among the renounced sensors capable of tracking the orientation and acceleration of the human body, Wearable inertial measurement units (IMUs) have proven to be among the best. It can monitor the orientation and acceleration of various human body parts by capturing movements without being hindered by object occlusions or clothing obstructions. Many researchers (*Huang et al., 2020b*; *Zhang et al., 2020*; *Liao et al., 2023a, 2023b*; *Zhao et al., 2023a*) proposed IMU-based pipelines to reconstruct 3D human poses to improve pose accuracy.

### Cloud and depth sensors

One of the hurdles encountered in 3D human pose estimation is the ambiguity related to depth. In recent years, there has been an increasing interest among computer vision researchers in employing depth sensors due to their precision and cost-effectiveness. Numerous studies have suggested using depth images for the estimation of 3D human poses (*Yu et al., 2018*; *Xiong et al., 2019*; *Zhou, Bhatnagar & Pons-Moll, 2020*; *Wang et al., 2023*). Additionally, research indicates that utilizing points can exhibit excellent performance in recovering 3D human mesh and other models of human pose (*Jiang, Cai & Zheng, 2019*; *Wang et al., 2020*; *Gu et al., 2022*; *Hermes, Bigalke & Heinrich, 2023*).

### Radio frequency sensors

Radiofrequency sensing devices have demonstrated an efficient and promising result in estimating 3D human poses. Researchers like *Zhao et al. (2019)*, and *Xie et al. (2023)* utilised radio frequency techniques in their work and promising output was obtained. The significant advantage of employing an RF-based sensing system lies in its capacity to move through walls and rebound off human bodies within the WiFi range without the need for carrying wireless transmitters. This approach provides a major benefit. Additionally, privacy can be maintained as the data is non-visual. However, it is worth noting that RF signals exhibit a relatively lower spatial resolution when compared to visual camera images, and RF systems have been demonstrated to yield coarse 3D pose estimations (*Zhao et al., 2018*; *Xie et al., 2022*, *2023*).

## Datasets for 3D human pose estimation

There are many datasets for 3D human pose estimation. For this article, only the most widely used deep learning–based 3D human pose estimation datasets are included.

**Human 3.6M** is the most widely used indoor dataset for 3D HPE from monocular images and videos. This dataset contains 3.6 million 3D human poses with 3D ground truth annotation captured by an accurate marker-based motion capture system. The training and testing images are categorized into subjects S1, S5, S6, and S7 for training, and images of subjects S9 and S11 for testing. Several researchers (*Reddy et al., 2021*; *Zhu et al., 2022*; *Chun, Park & Chang, 2023b*, *2023a*; *Shan et al., 2023*) utilised the Human 3.6M dataset for estimating the human pose.

**MuPoTS-3D** is another 3D dataset in which 3D poses were captured by a multi-view marker-less motion capture system containing 20 real-world scenes. This dataset contains many images with occlusions, drastic illumination changes, and lens flares. A total of 8,000 frames were collected in the 20 sequences by eight subjects. There are many more datasets for 3D human pose estimation that are not covered due to several reasons including page limitation. Many researchers utilised this dataset in their work. In another research performed by *Haque et al. (2016)*, *Sun et al. (2023)*, *Xing (2023)*, this dataset is also utilized.

**3D Poses in the Wild dataset (3DPW)** is the first unconstrained dataset in the wild with accurate 3D poses for evaluation. This dataset consists of about 60 video sequences, 3D body scans, and 3D object models out of which 18D models are different clothing. Among all the 3D benchmark datasets, 3DPW is reported to be the first dataset that contains video footage covered by a mobile phone camera. Its unique peculiarities inspired many researchers (*Cho et al., 2023*; *Ma et al., 2023*; *Nam et al., 2023*; *Oreshkin, 2023*; *Zhang et al., 2023b*) in the field of computer vision.

**MPI-INF-3DHP**, More than 1.3 million frames from 14 cameras were recorded in a green screen studio which allows automatic segmentation and augmentation. The dataset contains many human activities including walking, sitting, complex exercise poses, and dynamic actions. Other 3D HPE datasets consist of either outdoor or indoor scenes but MPI-INF-3DHP is reported to accommodate both complex outdoor and controlled indoor scenes respectively. This dataset is one of the top datasets utilized by many researchers

including (*Jiang et al., 2023*; *Mehraban, Adeli & Taati, 2023*; *Yu et al., 2023*; *Zhao et al., 2023b*).

## Evaluation metrics for 3D human pose estimation

Evaluating the performances of 3D human pose estimation is quite challenging because there are many features and equipment that need to be considered. Therefore, unlike like in 2D human pose estimation, there limited number of evaluation matric covered in this review.

### Mean per joint position error

It is the most frequently used metric to evaluate the performance of 3D HPE. Mean per joint position error (MPJPE) is calculated using the Euclidean distance between the projected 3D joints and the ground truth locations. Some researchers (*Güler, Neverova & Kokkinos, 2018*; *Wandt et al., 2021*) applied this metric to evaluate the performances of their models

### Mean average precision

This metric is used to measure the performance of predictions in the 3D dataset. Detection is successful when the predicted 3D body landmark falls within a distance less than the assigned mAP (mean average precision) value from the ground truth. The mAP is utilised as the evaluation metric to determine the unique rate of sample points (*Wang, Chen & Fu, 2022*).

### 3DPCK

It is a 3D extended version of the percentage of correct keypoints (PCK) metric used in 2D HPE evaluation. An estimated joint is said to be correct if the distance between the estimation and the ground truth is within a certain threshold. Generally, the threshold is set to 150 m. This metric is applied by *Pavlakos et al. (2019)* to evaluate the performance of their model.

### Euler angle error

It is a standard practice used to measure the error of 3D pose prediction in the Human 3.5 M dataset the evaluation is done by computing the Euclidean norm predictions and the ground truth Euler angle representations. The metric was applied in the Simulation experiments to verify the effectiveness of the proposed model (*Li et al., 2018*).

## RQ2: HOW ARE NEURAL NETWORKS APPLIED TO DIFFERENT 3D POSE ESTIMATION TASK

Research on 3B human pose estimation is gaining more popularity today. Many articles have been published in different journals and book chapters. Some of these articles published in Scopus are reviewed and summarized in this study as follows.

In *Yu et al. (2023)* an effective model known as Global-local Adaptive Graph Convolutional Network (GLA-GCN) was proposed to improve the 3D human pose lifting *via* ground truth data to improve the quality of estimated pose data. Three benchmark datasets were used in the experiments and the output shows that the proposed model

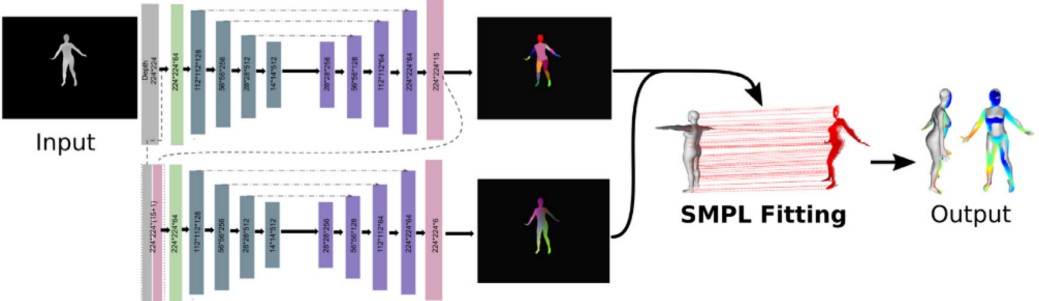

**Figure 3** The architecture of the proposed hybrid mode to predict 3D human pose and shape (*Wang et al., 2023*). Direct License. https://s100.copyright.com/CustomerAdmin/PLF.jsp?ref=cef4c52c-cb41-4996-9ed0-5b42de22dfc5.

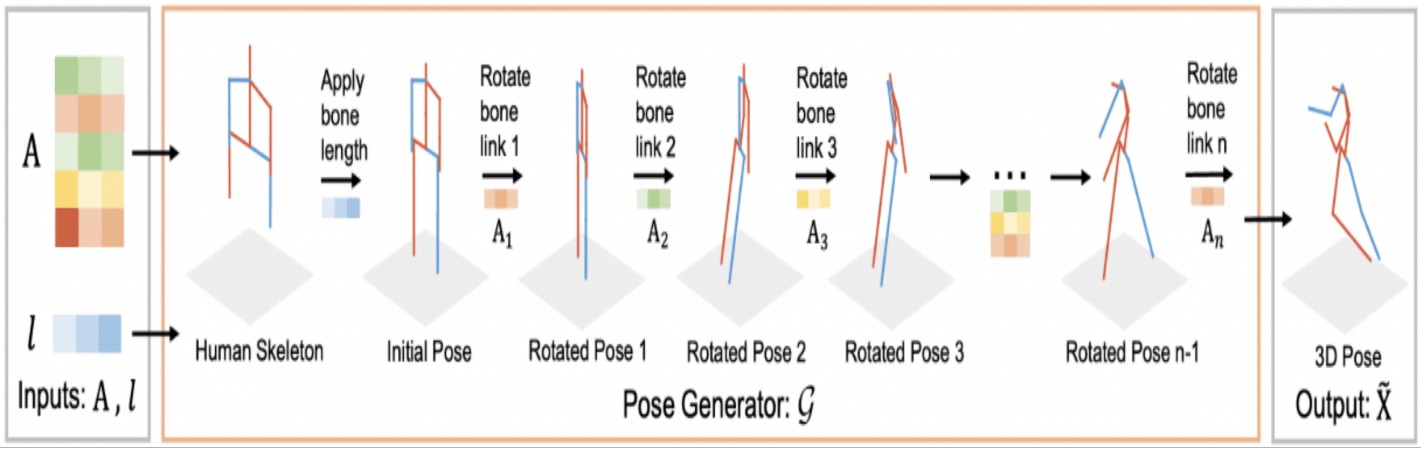

**Figure 4** 3D human pose generator (*Guan et al., 2023*).

performed better than the state-of-the-art model. A spatial-temporal mesh attention convolution (MAC) was developed (*Wang et al., 2020*) to predict the 3D coordinates of mesh vertices at the high resolution based on the estimated 3D coordinates and features at the low resolution. The framework was generalised to the real data of human bodies with a weakly supervised fine-tuning method. The developed model achieves higher accuracy in recovering the 3D body model sequence from a sequence of point clouds.

To tackle the shortcomings of estimating the pose and shape of human bodies (*Wang et al., 2023*) developed a hybrid pipeline that put together the strength of both DL-based and sensor-based models. After performing the experiments with four benchmark datasets which include the DFAUST dataset, SURREAL, and some parts of AMASS dataset, the outcome of the experiments shows that this hybrid approach enables us to enhance pose and shape estimation compared to using DL or model fitting separately. The hybrid model is presented in Fig. 3.

A novel human pose generator was developed (*Guan et al., 2023*) to generate diverse 3D human poses. The performance of the proposed model known as PoseGU was evaluated

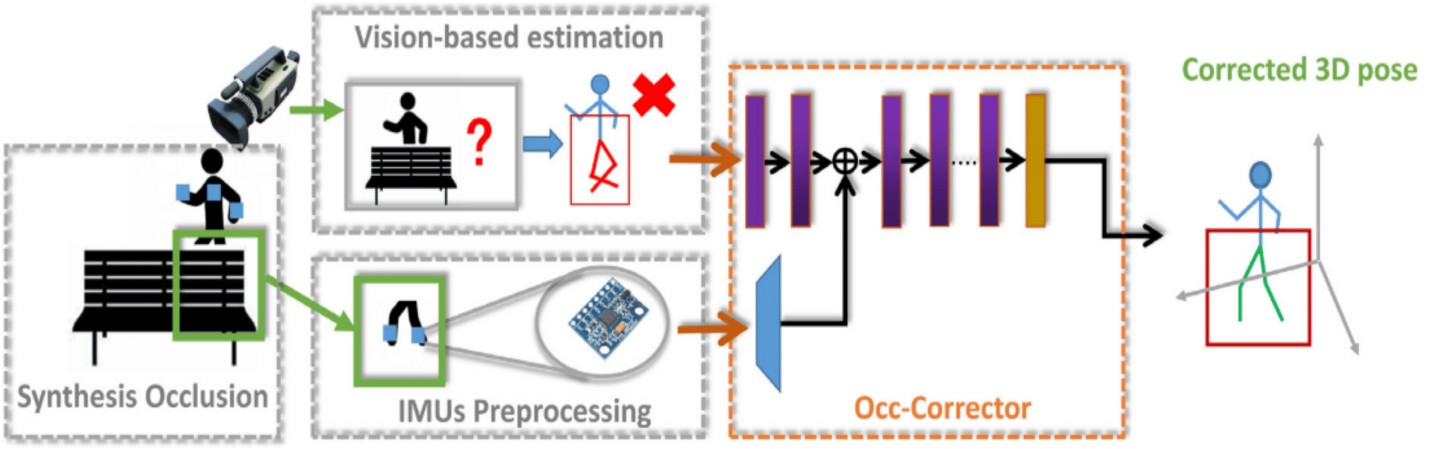

**Figure 5 Occ-Corrector framework (*Zhao et al., 2023a*).**

with three different benchmark datasets and the experimental analysis shows that it generates 3D poses with improved data diversity and better generalization ability. The framework of the developed module is shown in Fig. 4.

Estimating the 3D pose of humans in a clinical setting requires a simple and accurate system. The current method is attached to many devices which makes it difficult in a clinical environment. A model that attempts to reduce the number of devices and maintain its accuracy is developed by *Zhao et al. (2023a)*. A semantic convolution-based neural network known as Occ-Corrector is developed to deal with occlusion issues that might arise during the estimation of 3D poses from a single camera by integrating with IMU's sensors. Occ-Corrector is presented in Fig. 5.

The summary of the current deep learning-based article from Scopus is shown in Table 1. Table 1, summarizes the literature of 30 of the most recently published articles in Scopus, illustrating the Study, the model utilized, the type of dataset and a summary of the aim achievement made and the limitation of each study.

### RQ3 WHAT AREAS OF HUMAN ENDEAVOUR WHERE HUMAN POSE ESTIMATION IS APPLICABLE

## Area of application

In terms of real-world applications, 3D human pose estimation provides numerous applications in human endeavours. Here some popular applications of 3D HPE are reviewed and presented.

### *Business activities*

Human pose estimation has proven valuable in both real-world and virtual business settings. The influence of e-commerce trends, particularly in areas like face masks and clothing purchases, has been noteworthy. Traditional depictions of clothing items in

images are no longer sufficient to meet customer expectations. Consumers now desire a more dependable representation, wanting to visualize how selected clothes will look on them. Through 3D human pose estimation (3D HPE), it becomes feasible to generate realistic representations of human body regions for virtual fitting rooms, specifically for clothes and face mask analysis. This is achieved through techniques such as clothes parsing (*Saito et al., 2019*; *Yu et al., 2019*) and pose transfer (*Li, Huang & Loy, 2019*), which infer the three-dimensional appearance of an individual wearing specific garments.

### Entertainment

In the gaming, film, and animation sectors, 3D HPE plays a pivotal role. While motion capture systems serve as the foundation for these industries, addressing the intricate movements of actors, 3D HPE is increasingly employed as a cost-effective alternative to high-priced motion capture devices. With the help of 3D pose estimation and human mesh recovery, 3D character animation from a single photo is developed by *Weng, Curless & Kemelmacher-Shlizerman (2019)*.

### Healthcare

Human pose estimation finds practical application in the fields of human rehabilitation and physiotherapy. It involves tracking human activities during rehabilitation exercises by precisely identifying key points of the patient's movements for effective treatment. *Gu et al. (2019)* proposed a physiotherapy system designed to assess and guide patients at home. Additionally, *Weiming Chen, Guo & Ni (2020)* utilized HPE algorithms for monitoring fall detection, enabling prompt assistance. Consequently, 3D pose estimation techniques have the potential to develop a system for correcting sitting postures. Such a system could monitor user status, provide quantitative human motion information, aid in diagnosing complex diseases, formulate rehabilitation training, and facilitate physical therapy under the guidance of physicians.

### Sport activities

In sports, coaches and trainers analyse and monitor athletes' performance. The advancements in 3D HPE have enabled Artificial Intelligence (AI) trainers to provide precise coaching through action detection techniques using just a few camera settings. *Hwang, Park & Kwak (2017)* developed a system, leveraging 3D poses extracted from videos, for performance analysis, rapid response, and improvement. *Zecha et al. (2018)* employed human pose estimation techniques to precisely track and assess athletes' performance in swimming exercises, ensuring accurate metrics at any given time. *Wang et al. (2019)* developed an AI coaching system incorporating a pose estimation module, offering personalised assistance for athletic training.

### Security/safety

3D human pose estimation finds application in surveillance, involving the estimation and analysis of a person's poses or activities in a surveillance environment. Many advanced shopping malls and stores have implemented cashier-less systems to identify customers

engaging in suspicious behaviour. These establishments aim to identify any irregularities among their customers by employing a hybrid computer vision system that combines camera sensor networks and Internet of Things (IoT) devices with HPE. Human pose estimation plays a crucial role in scenarios where the actual contact between the customer and the product is not visible to the camera. In such cases, the HPE model analyses the positions of customers' hands and heads to determine whether they have taken a product from the shelf or left it in place. This system utilizes pose information for action recognition, tracking, prediction, and detection. Researchers like *Angelini et al. (2018)* have proposed real-time action recognition methods using pose-based algorithms, while human action detection in videos has also been explored (*Cao et al., 2020*).

### RQ4: WHAT ARE THE CURRENT CHALLENGES OF 3D HPE

## Challenges

No doubt about how powerful deep leaning-based models are, in dealing with different types of computer vision problems specifically human pose estimation. Unlike 2D HPE, 3D HPE models are facing different challenges in the implementation of deep learning–based pipelines.

### *Occlusion*

Is one of the main challenges facing the estimation of 3D human pose in a deep learning-based approach. The act of occluding the target object is done by covering the required key points of that object which automatically affects the accuracy of 3D pose estimation. Occlusion may be self-occlusion, inter-person-occlusion, out-of-frame occlusion, *etc*.

### *Inadequate data*

One of the requirements of any deep learning model to enable accurate learning is the availability of data. Researchers in this field are facing the serious challenge of limited data with limited poses, inadequate pose variations, and inadequate annotation. Deep learning models are data-driven models that require large data with different poses for training and calibration. Thus inadequate data will affect the performance of any deep learning model.

### *Tampered input data*

Data that is being tampered with may affect the accuracy of estimation despite the efficiency of the deep learning model. Some data are tampered with by adding noise, blurring, low or high contrast, and resolution which seriously affect the estimation accuracy.

### *Complex scene*

The complexity of an image particularly in multi-person pose estimation can affect the 2D pose estimation and also the 3D pose estimation. When the 2D estimation is not accurate, estimating the 3D can be affected since the 3D pipeline estimates the 2D key points first before estimating the 3D joints. An example of this is heavy crowd, football games and so on.

## RQ5: WHAT ARE THE FUTURE RESEARCH DIRECTIONS FOR 3D HUMAN POSE ESTIMATION

### Multi-person reconstruction

In every community setting, individuals frequently engage in activities such as walking, talking, or collaborating in groups, such as families or teams. A compelling avenue for future exploration involves reconstructing groups of people across both spatial and temporal dimensions, thereby unveiling relationships and activities within the targeted group. Additionally, when addressing person matching across various cameras or extended temporal sequences, leveraging the relationships among individuals within a group offers a more stable context that can be utilized to address challenges like occlusions or detection failures. This task can also be integrated with person tracking (*Rajasegaran et al., 2021*) and re-identification (*Lisanti et al., 2017*) to enhance the robustness of reconstruction, especially in crowded scenarios.

### Physical constraints

Many current approaches overlook the interaction between humans and 3D scenes. There exist significant constraints in the relationship between humans and scenes, such as the inability of a human subject to occupy the same locations as other objects in the scene simultaneously. By exploring these physical constraints alongside semantic cues, it is possible to enhance the reliability and realism of 3D HPE.

### Full body reconstruction

Parametric models such as SMPL and SMPL-X are limited to representing individuals with minimal clothing. To surpass the representational limitations of parametric models, the research community must explore alternative models that offer greater flexibility. Previous studies have employed meshes, point clouds (*Ma et al., 2021*), and implicit fields (*Li et al., 2020b*) to capture detailed clothing deformation. While these approaches can yield reasonable outcomes, the reconstructed surfaces often appear overly smoothed and lack robustness when confronted with novel poses. Addressing these issues involves incorporating diverse types of representations (*Shao et al., 2022*) to harness the modeling capabilities of varied approaches.

### Adaptation of HPE domain

In certain scenarios, like the estimation of human pose from images of infants (*Huang et al., 2021*) or collections of artwork (*Madhu et al., 2022*), there is a scarcity of training data accompanied by accurate ground truth annotations. Additionally, the data for these applications showcase distributions distinct from those found in standard pose datasets. HPE methods trained on conventional datasets might struggle to generalize effectively across diverse domains. A recent approach to mitigate this domain gap involves the use of generative adversarial network (GAN)-based learning techniques. However, the effective transfer of human pose knowledge to bridge these domain gaps remains an unexplored challenge.

### Generalized metrics

3D human pose estimation finds applications in visual tracking and analysis. Current methods for reconstructing 3D human pose and shape from videos lack smooth and continuous results. One contributing factor is that evaluation metrics like MPJPE do not assess smoothness and the level of realism adequately. There is a need to develop suitable frame-level evaluation metrics that specifically address temporal consistency and motion smoothness.

## CONCLUSIONS

This article explores recent advancements in deep learning-based 3D human pose estimation models. First, we provide a brief introduction to human pose estimation and its various types, followed by an in-depth discussion of the evolution, and achievements of these models. Second, we review studies that apply 3D pose estimation models to different tasks, such as hand pose, full-body pose, and human activities, highlighting the main challenges in the field. Third, we explain the diverse applications of 3D human pose estimation across various domains and list all potential neural network modules for addressing 3D HPE problems. Fourth, we offer readers a comprehensive understanding of existing approaches and outline detailed future directions for deep learning-based 3D human pose estimation. Lastly, our review aims to serve as a comprehensive guide for both industry and academic practitioners, providing a significant and rich understanding of various aspects of estimating three-dimensional human poses while advocating for continued advancements in the field.

## ACKNOWLEDGEMENTS

We express our genuine gratitude to everyone who offered their support and assistance throughout the research endeavour, contributing to the successful completion of this survey.

### Funding

This research is funded by the Universiti Teknologi PETRONAS through a Short-Term Internal Research Funding (STIRF) Grant (Grant Number: 015LA0-057) (previously known as Cost Centre: YUTP-FRG 015LC0-373). The funders had no role in study design, data collection and analysis, decision to publish, or preparation of the manuscript.

### Grant Disclosures

The following grant information was disclosed by the authors:
Universiti Teknologi PETRONAS through a Short-Term Internal Research Funding (STIRF): 015LA0-057.
Previously known as Cost Centre: YUTP-FRG 015LC0-373.

## Competing Interests

The authors declared that there is no competing interest.

## Author Contributions

- Sani Salisu conceived and designed the experiments, performed the experiments, analyzed the data, prepared figures and/or tables, authored or reviewed drafts of the article, and approved the final draft.
- Kamaluddeen Usman Danyaro conceived and designed the experiments, performed the experiments, analyzed the data, authored or reviewed drafts of the article, and approved the final draft.
- Maged Nasser conceived and designed the experiments, analyzed the data, prepared figures and/or tables, authored or reviewed drafts of the article, and approved the final draft.
- Israa M. Hayder performed the computation work, prepared figures and/or tables, and approved the final draft.
- Hussain A. Younis conceived and designed the experiments, performed the experiments, performed the computation work, prepared figures and/or tables, authored or reviewed drafts of the article, and approved the final draft.

## Data Availability

The article is a literature review.

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
