# Peer review of "Review of models for estimating 3D human pose using deep learning"

_PeerJ Computer Science, doi:10.7717/peerj-cs.2574_

## Round 0.1 · original submission · Major Revisions

Please consider the reviewer comments.

Reviewer 1 ·

Basic reporting

English could be improved
The field has several new articles published recently; The contribution needs to be clear in this matter.

Experimental design

The authors didn't present the research questions that they are pursuing...

The number of papers obtained from searches is not disclosed, nor the duplicated ones, nor the rejection criteria:
line 101 - "Subsequently, the full text of screened papers was meticulously examined to ascertain their relevance for inclusion or exclusion.";
line 105 - "(...) rejection criteria were applied."

Authors mention in line 195: "These criteria stipulated that articles must be original reviews or surveys, authored in English (...)". Is the manuscript built from searches or previous published reviews?

Validity of the findings

My primary concern is to extract the impact and novelty provided by this manuscript. In contrast, in a quick search on Google Scholar, and only for 2023 and further, there are a lot of reviews for this specific problem. In what way is yours a clear and relevant contribution? The search: (https://scholar.google.com/scholar?as_ylo=2023&q=3d+human+pose+estimation+reviews&hl=en&as_sdt=0,5); Because that actually the research gap that the authors claim to address (line 71) is for "2D HPE (...) while survey and literature reviews on 3D models HPE are lacking."

The conclusion must be improved, as the research questions are not responded to (or even presented)

Additional comments

I would advise a reformulation of the manuscript in order to make it a systematic review, following some survey methodology, like PRISMA, as an example.

Cite this review as

·

Basic reporting

Strengths:
1. This review paper focuses on the deep-learning-based 3D human pose estimation, and broadly covers recent papers in related fields.

Weaknesses:
1. According to the description in Lines.71-78, authors point out that the survey about 3D HPE based on deep learning is lacking, but it is easily found some other recent review papers on this topic, including but not limited to:
• Zheng C, Wu W, Chen C, Yang T, Zhu S, Shen J, Kehtarnavaz N, Shah M. Deep learning-based human pose estimation: A survey. ACM Computing Surveys. 2023 Aug 26;56(1):1-37.
• Wang J, Tan S, Zhen X, Xu S, Zheng F, He Z, Shao L. Deep 3D human pose estimation: A review. Computer Vision and Image Understanding. 2021 Sep 1;210:103225.
There’s no obvious difference claimed in this paper compared with these highly related review papers.
2. Figures and tables are supposed to appear at the appropriate locations in the main body of the paper rather than being piled at the end. Besides, the figures should be scaled accordingly, since the texts in current figures are either distorted or too small.
3. The writing should be improved. There’re typos, grammar errors and duplications in the current version, e.g.:
• Lines.35-37, “this study explored … are explored”
• Lines.154-156, “Several researchers … Several researchers …”
• Line.668, “Thirty 30 of the most recently published articles …”

Experimental design

Strengths:
1. The survey is conducted based on reasonable search methods among different databases.
2. Besides different methods, commonly used datasets and evaluation metrics, authors also conclude the current challenges of 3D human pose estimation.

Weaknesses:
1. The organization of the paper is confusing:
• Firstly, there is no hierarchical structure of the paper, all the contents use the same-level section titles.
• Secondly, the division of paragraphs is fragmented, especially in the Introduction.
• Thirdly, the description of this paper is not well organized. After introducing modeling and methods, authors discuss the applications and go back to methods again.
2. Although this paper focuses on the 3D pose estimation, authors also include many 2D estimation methods without clear demonstration. Among traditional methods given in Lines.51-58, many of them are not even for 3D cases according to given references, e.g. the silhouette contours, pictorial structures, etc.
3. When introducing the evaluation metrics, corresponding citations should be given.
4. There’re mismatches between descriptions and citations, e.g. in Line.285, “He et al. (Xiao et al., 2023) …”, I couldn’t find such a paper in the reference list whose authors contain both “He” and “Xiao”.

Validity of the findings

Strengths:
1. Sufficient papers are mentioned to cover recent achievements on 3D human pose estimation, which meets the goal set in the Introduction.
2. Authors give detailed discussion about future directions in the Conclusion.

Weaknesses:
1. As a review paper, more details about related works should be provided. However, most recent works are only concluded in a table with their names, datasets aims and achievements. Authors should describe the main ideas of their methods in brief.
2. From Figures.8-13, the figures cannot fully reflect the models discussed in this review paper. When introducing different models for 3D HPE, authors should use architectures of the most representative corresponding methods instead of using a general model structure with totally different input and output requirements.
3. There’re also some issues in authors’ summary:
• In Lines.193-195, when introducing the fundamental stages in 3D HPE, authors write that the estimation of 3D poses based on 2D key points is achieved by combining 2D frames, but for single-image case, there’s no way to get the temporal information.
• Starting from Line.206, authors introduce 3D HPE from images and videos. I suggest that authors separately discuss these two sources. Compared with single image, videos can provide temporal information, which can help solve occlusion problems to some degree, even without multi-view setting.

---

## Round 0.2 · Major Revisions

Based on the reviewers comments, the authors did not appropriately address the concerns of Reviewer 1 (or they did not integrate their responses properly into the revision). You I am giving you a last chance to improve the manuscript for its acceptance.

Reviewer 1 ·

Basic reporting

Check below.

Experimental design

Check below.

Validity of the findings

Check below.

Additional comments

The authors indeed improved part of the paper.
Nevertheless, when reading the rebuttal letter (RL) and checking the changes in the manuscript, things don't match every time. So:
1) The abstract doesn't have a summary of your main findings
2) Did you use review papers as an inclusion criterion? Why? In what sense is yours better or improved in this scientific area?
3) Figure 1 is very blurred.
4) The authors mention in RL that they were pursuing 5 research questions. I can't find those questions.
5) The rejection criteria is not mentioned in the text.
6) The authors mention in RL that they follow the PRISMA methodology. I'm not sure if this. Please check https://www.prisma-statement.org/ and substitute Figure 1.
7) I can't find Table 1.
8) The list of references was not updated.

Cite this review as

·

Basic reporting

I appreciate authors’ efforts on improving the quality of their manuscripts. Due to the inconsistency between the final pdf for review and the word document with marks, the following reviews are mainly based on the submitted pdf file. In general, all my previous concerns have been solved.
1. In the current version, authors have emphasized the differences between their work and other review papers. Authors also have added more details about specific 3D HPE methods in the latest submission.
2. The current manuscript has a clearer organization with titles in different font sizes. If possible, I suggest that authors could also add numerical orders to the section titles, such as “1. INTRODUCTION”, “3.1 Research questions”, to make them more distinguishable.
3. Missing citations for evaluations metrics have already been added.
4. Previous typos and figure problems have been corrected.

Experimental design

No comment.

Validity of the findings

No comment.

---

## Round 0.3 · Minor Revisions

Based on the reviewer comments, the manuscript must be revised accordingly.

Reviewer 1 ·

Basic reporting

See below

Experimental design

See below

Validity of the findings

See below

Additional comments

Dear authors,
In fact, the manuscript has improved considerably, although it has some issues that remain:
Until section 4, the study presents a conventional structure. So, the authors present the bibliographic review and the search results, considering the various inclusion and exclusion criteria.
The issue is in the transition to chapter 5, which seems out of place to me in the sense of not being in a logical sequence with what has been presented so far.
I would expect chapter 4 to have a description and critical evaluation of all 30 contributions that passed the filtering carried out. At the end of these descriptions, it would make sense to have Table 1, which could also include the advantages and limitations of each study. In this sequence, answering the various RQs would make sense.
My suggestion would be:
- Dilute the current cap2 in cap 1;
- Move the current chapter 5 to the location of the current chapter 2, as the explanation of the various representations must be prior;
- The current chapters 6 and following are not numbered chapters. They must be subsections and follow the current information in Chapter 4. Please consider the previous comment before this change.

Cite this review as

---

## Round 0.4 · accepted · Accept

Based on the reviewer comments, the manuscript can be accepted.

Reviewer 1 ·

Basic reporting

All the questions have been appropriately addressed.

Experimental design

All the questions have been appropriately addressed.

Validity of the findings

All the questions have been appropriately addressed.

Additional comments

All the questions have been appropriately addressed.

Cite this review as